# Peptidomic Analysis on Mouse Lung Tissue Reveals AGDP as a Potential Bioactive Peptide against Pseudorabies Virus Infection

**DOI:** 10.3390/ijms23063306

**Published:** 2022-03-18

**Authors:** Yijie Ma, Shimao Tian, Qianhui Wan, Yingying Kong, Chang Liu, Ke Tian, Hongya Ning, Xiaodong Xu, Baomin Qi, Guihong Yang

**Affiliations:** 1Key Laboratory of Animal Pathogen Infection and Immunology of Fujian Province, College of Animal Sciences (College of Bee Science), Fujian Agricultural and Forestry University, Fuzhou 350002, China; mayijie@fafu.edu.cn (Y.M.); tianshimao@fafu.edu.cn (S.T.); wanqianhui@fafu.edu.cn (Q.W.); kongyingying@fafu.edu.cn (Y.K.); 3185503082@m.fafu.edu.cn (C.L.); ninghongya@fafu.edu.cn (H.N.); xuxiaodong@fafu.edu.cn (X.X.); 000q040038@fafau.edu.cn (B.Q.); 2College of JIN SHAN, Fujian Agricultural and Forestry University, Fuzhou 350002, China; 186726050@m.fafu.edu.cn

**Keywords:** Pseudorabies virus, peptidomic analysis, AGDP, inflammation, virus replication, cytokines

## Abstract

Pseudorabies virus (PRV) infection could cause severe histopathological damage via releasing multiple factors, including cytokines, peptides, etc. Here, peptidomic results showed that 129 peptides were identified in PRV-infected mouse lungs and were highly involved in the process of PRV infection. The role of one down-regulated biological peptide (designated as AGDP) during PRV infection was investigated. To verify the expression profiles of AGDP in response to PRV infection, the expression level of the precursor protein of AGDP mRNA was significantly decreased in PRV-infected mouse lungs and cells. The synthesized AGDP-treating cells were less susceptible to PRV challenges than the controls, as demonstrated by the decreased virus production and gE expression. AGDP not only inhibited the expression of TNF-α and IL-8 but also appeared to suppress the extracellular release of high-mobility group box 1 (HMGB1) by inhibiting the output of nuclear HMGB1 in cells. AGDP could also inhibit the degradation of IκBα and the phosphorylation levels of P65 after PRV infection. In total, our results revealed many meaningful peptides involved in PRV infection, thereby enhancing the current understanding of the host response to PRV infection, and how AGDP may serve as a promising candidate for developing novel anti-PRV drugs.

## 1. Introduction

Pseudorabies virus (PRV) belongs to the family of the *Herpesviridae*, subfamily *Alphaherpesvirinae*, genus Varicellovirus [1]. PRV is a double-stranded DNA genome of approximately 150 kb encoding at least 100 proteins, such as gB, gC, gD, gE, etc. [2]. PRV can infect various mammals, including pigs, cattle, goats, sheep, dogs, rabbits, cats, rodents, wild animals, etc. [3,4,5,6]. Pigs are the only natural host of PRV, and infection can lead to pulmonary edema and interstitial pneumonia, with histopathological observations showing alveolar septa thickening and lymphocytic infiltration [7]. Traditionally, the most effective measure for preventing and controlling PRV infection is annual vaccination. However, due to the continuous variation of PRV, the current commercial vaccines have been unable to control PRV infection in China since 2011 [8,9]. Specifically, PRV continues to evolve, with new antigenic variants emerging in humans in 2018, who display acute encephalitis, neurological disorders, severe visual impairment, and pulmonary inflammation [10]. This indicated that traditional vaccines were considerably less efficacious than predicated. Thus, it is urgently necessary to develop novel effective measures to curb PRV in pigs in China, and developing alternatives to annual vaccines is the most promising measure to control this disease.

The host’s innate immune system provides the first line of defense against viral infection [11]. Several studies have demonstrated that the underlying interactions between PRV and the host’s innate immune system are mainly mediated through several signaling pathways, such as type I interferon (IFN) and NF-κB signaling pathways, and modulates the expression of proinflammatory factors [12]. On one hand, the host’s innate immune response is activated by the production of IFNs after viral infection, thus further activating a positive feedback loop by binding to the IFN-α/β receptor (INFAR) in an auto- or paracrine manner [11]. The type I IFN/IFAR axis could trigger a major innate immune response against the viral infection [13]. This evidence demonstrated that the type I IFNs/INFAR signaling pathway is required for host defense against PRV infection [14]. On the other hand, PRV infection triggered the activation of the NF-κB signaling pathway [12]. NF-κB signaling can be activated by various stimulus factors and has thus typically resulted in a robust surge of several proinflammatory factors, along with negative feedback factors to inhibit excessive inflammation [15,16]. Additionally, two recent studies have addressed the fact that PRV infection could trigger a non-canonical NF-κB pathway independent of IKKβ kinase activity, which would not result in the expression of tumor necrosis factor alpha (TNF-α) or interleukin-6 (IL-6), and not induce the negative-feedback-loop genes, such as IκBα [17,18]. These studies indicated that PRV could efficiently inhibit the potentially antiviral response of hosts by triggering the activation of a non-canonical DDR-NF-κB signaling axis, but these results could not exclude the other potential signaling pathways. Therefore, it is important to further understand the associations between PRV and the innate immune system for developing novel anti-PRV infection measures.

Peptides are important members of the host’s innate immune system, which was recently highlighted in the process of micro-pathogens’ stimuli and showed different abundance levels [19,20,21]. With the advent of peptidomics, therapeutic peptides have made great progress in the field of drug development [22], such as anti-tumor [23], antiviral [24,25], antimicrobial [26], and immune-regulation drugs [27]. The expression levels of the peptides were always altered under different pathological environments, e.g., down-regulated expression of LL-37 was found in tumors compared to normal controls [28], and up-regulated NMB expression was found in response to influenza A virus (IAV) infection and thereby contributed to the host’s anti-IAV responses [29]. These reports suggest that the differentially expressed peptides (DEPs) may play specific functions under various stimuli, and these DEPs could always act as the important targets for drug design. However, DEPs and their potential functions in responding to PRV infection remain unclear and need further study. This study used peptidomics to analyze the significant DEPs in mouse lungs between the PRV-infected and control groups, combined with Gene Ontology (GO) and the Kyoto Encyclopedia of Genes and Genomes (KEGG) analysis, to discover new therapeutic peptides. This study’s results showed that a total of 129 DEPs were involved in the process of PRV infection. Here, we extend the current findings on the peptidomic profile by characterizing the potential role of AGDP in responding to PRV infection, which provides new insight into the search for sensitive and non-invasive biological markers and presents novel clues for the design and development of more effective therapeutic strategies for PRV infection.

## 2. Results

### 2.1. Establishment of PRV Infection Models in Mice

To confirm whether the mice have been successfully infected with PRV, the lung tissues collected from the two groups were analyzed by H&E, PCR, and WB, respectively. Compared to the control group, PRV-infected lungs were observed for typical clinical symptoms of histopathological damages caused by H&E staining, which have the characteristics of consolidation of tissue, internal hemorrhage of tissue, and markedly enlarged alveolar septa (Figure 1A). The results of PCR and WB consistently displayed that the virus gE was only expressed in PRV-infected lungs and not in the control group (Figure 1B,C). These results indicated that the above two groups were successfully established and could be used effectively for the subsequent peptidomic analysis.

### 2.2. Identification of Bioactive Peptides and Bioinformatics Analysis

A powerful iTRAQ-labeled quantitative peptidomic analysis based on LC-MS/MS was conducted to profile the mouse lung peptidome between PRV-infected groups and control groups. In total, DEPs were identified and quantified from the two groups when the iTRAQ fold-change ratios were ≥1.5 or ≤0.67, while the *p*-valued was ≤0.05. Principal component analysis indicates the two groups had a reproducible result (Figure 2A). Through the analysis of the significant DEPs, we found that these peptides were mainly distributed in the range of the length of 8 to 25 amino acids (Figure 2B), among which 129 peptides were significantly differentially expressed, with 46 up-regulated and 83 down-regulated (Figure 2C,D). The cluster heat map reveals the distribution of the top 25 up-regulated peptides and the top 25 down-regulated peptides (Figure 2E,F).

To explore the potential physiological functions of the DEPs, GO annotation was performed on the precursor proteins of the DEPs. As shown in Figure 3, GO annotation includes three parts: the biological process, cellular component, and molecular function. Biological process annotations mainly include the cellular process, the developmental process, binding, the cellular anatomical entity, the metabolic process, the multicellular organismal process, the response to stimulus, localization, catalytic activity, and the immune system process (Figure 3A). Cellular component annotations include the cellular anatomical entity and protein-containing complex (Figure 3B). Molecular function annotations mainly include the cellular process, binding, the cellular anatomical entity, and catalytic activity (Figure 3C). To further understand the biological functions of differential peptide precursor proteins, we used the KEGG database to identify the main biochemicals’ metabolic pathways and signal transduction pathways involved in the precursor proteins of DEPs (Figure 4), which includes the regulation of the actin cytoskeleton, phagosome, leukocyte transendothelial migration, focal adhesion, tight junction, systemic lupus erythematosus, etc.

The sequence of the RAGE-derived peptide (AGDP) was distributed between the 309 and 329 amino acids of the RAGE protein (Figure 5A). The interaction networks of RAGE and its associated proteins were analyzed according to STRING (Version 11.5, http://string-db.org/, accessed on 15 December 2021). The results indicated that the main interaction networks existed with proteins HMGB1, Rela (P65), NFκB1, and NFκB2 (Figure 5B).

### 2.3. Down-Regulated Expression of RAGE mRNA in the Response to PRV Infection

To analyze the potential roles of AGDP involvement in PRV infection, we further detected the expression levels of RAGE mRNA in mouse lung tissues at 3 dpi and MLE-12 cells at 24 hpi by RT-PCR and qRT-PCR. The results demonstrated that the expression levels of RAGE mRNA were significantly decreased in mouse lungs at 3 dpi (Figure 6A,B) and in MLE-12 cells at 24 hpi (Figure 6C,D).

### 2.4. AGDP Exhibited Anti-PRV Infection and Inhibited the Inflammatory Response In Vitro

To explore the potential roles of AGDP in response to PRV infection, the expression levels of TNF-α, IL-8, gE, and the virus titer in the supernatants of MLE-12 cells were detected, respectively. The results of RT-PCR/PCR showed that the expression levels of TNF-α, IL-8, and gE at 24 hpi were down-regulated by AGDP treatment (Figure 7A), and the more significant results were obtained from qRT-PCR and qPCR assays (Figure 7B–D). Substantially, the results showed that AGDP treatment could decrease the expression levels of gE protein at 24 hpi with WB (Figure 7E). Consistently, the plaque assay results demonstrated that AGDP could significantly decrease the virus production (Figure 7F). Thus, these results indicated that AGDP could exert anti-PRV infection properties by inhibiting the expression of these inflammatory cytokines and the viral production in vitro.

### 2.5. AGDP Suppressed the Extracellular Release of HMGB1 from the PRV-Infected MLE-12 Cells

To clarify the anti-inflammation properties of AGDP during PRV infection, the expression levels of HMGB1 in MLE-12 cells were firstly investigated by RT-PCR, qRT-PCR, and WB, respectively. The results showed that AGDP treatment could significantly down-regulate the expression levels of HMGB1 mRNA in PRV-infected conditions (Figure 8A,B). Meanwhile, the cellular localization of the HMGB1 protein in MLE-12 cells was observed by IFC. The results indicated that the main localization of the HMGB1 protein was observed in the nucleus of MLE-12 cells, and PRV infection could induce the output of the nucleus HMGB1 protein. However, the output of the nuclear HMGB1 protein in MLE-12 cells could be further suppressed by AGDP treatment following PRV infection (Figure 8C). Interestingly, the expression levels of the HMGB1 proteins were significantly decreased in MLE-12 cells supernatants, but not in the cell lysis (Figure 8D).

### 2.6. AGDP Regulated the Degradation of IκBα Protein and the Phosphorylation Levels of P65 Protein in PRV-Infected Cells

To better understand the anti-inflammatory function of AGDP during PRV infection, the regulatory effects of AGDP on the activities of IκBα and P65 proteins were measured in MLE-12 cells by WB. The results showed that AGDP could suppress the degradation of IκBα protein and inhibit the phosphorylation levels of P65 protein in response to PRV infection (Figure 8E). These results further validated the regulation of AGDP in the inflammatory cytokines responding to PRV infection.

## 3. Discussion

PRV is still a major public health concern, seriously affecting animals’ lung tissues and nervous systems, and even posing a threat to human health [30]. Although significant progress has been made in the past decades, understanding the pathophysiological process and facilitating the control of this disease is still needed. However, the differences in peptide levels and their roles during PRV infection remain poorly understood. Thus, peptidomic analysis of lung tissues derived from PRV-infected animals is appealing. In this study, 129 DEPs were identified based on peptidomic analysis, when the animals were successfully infected with PRV. These results indicated that meaningful peptides might participate in the pathogenesis of PRV infection. Knowledge obtained from the characteristics of these DEPs in PRV-infected lungs might provide new insights in an in-depth exploration of the pathogenesis of PRV and novel therapeutic strategies for this disease.

Based on the information gathered using the database, we attempted to ascertain the activities of one peptide in response to PRV infection. This peptide was derived from the C2 domain region of RAGE [31], and was designated as AGDP. Since RAGE plays a key role in the regulation of immune system responses, and inflammatory processes, as well as various types of cancers [32], it could mediate the cellular progress and binding and correlate with the pathways of Jak/STATs and NF-κB signaling, which are considered promising targets for the development of novel anti-inflammatory diseases [33,34]. These findings indicated that AGDP might play an important regulatory role during PRV infection. Increased RAGE expression has been indicated in many acute and chronic inflammatory events, while the decreased RAGE expression was also observed in pulmonary samples [35]. In the present study, the expression of RAGE mRNA was significantly decreased in PRV-infected MLE-12 cells, which was consistent with the above results of peptidomics. Although the mechanism for down-regulation of RAGE expression in lungs remains unclear, RAGE is constitutively or inducibly expressed depending on the cell types and the pathological process. According to the concept that the down-regulated levels of RAGE drive the anti-inflammatory function in inflammatory conditions, we thus speculated that the down-regulated expression of AGDP in peptide levels induced by PRV infection may play a role in anti-inflammatory responses.

To clarify the role of AGDP involvement in PRV infection, synthesized AGDP was employed to stimulate the cells undergoing PRV challenges. PRV infection could induce the robust release of several proinflammatory cytokines to cause severe inflammatory damage to the lungs [36,37]. Here, it was observed that the expression of proinflammatory cytokines IL-8 and TNF-α and virus gE induction by PRV infection were substantially inhibited by AGDP treatment. These results indicated the promising effects of AGDP playing an anti-inflammation function in responding to PRV infection. The following virus titration displayed the significant inhibitory effects of AGDP in the viral production, which is a key point for PRV infection, further highlighting the anti-inflammation function of AGDP in responding to PRV infection. Together, the above results indicated that AGDP could exert an anti-inflammatory function induced by PRV infection. RAGE activities have been presented in several physiological and pathological processes by interacting with a diverse class of ligands, including high mobility group box1 (HMGB1) and S100, etc. [38]. In particular, HMGB1 is associated with many disease, such as sepsis [39], influenza virus [40], and cancer [41], which could interact with the extracellular region of RAGE and mediate the inflammatory response [34]. Although AGDP shares the same structure of the extracellular C2 domain region of RAGE, whether HMGB1 could also mediate AGDP playing an anti-inflammatory function while challenged by PRV remains to be further studied. To gain a better understanding of the protective role of ADGP during PRV infection, we further investigated the regulation of ADGP on the expression of HMGB1. After addition with AGDP, the expression of the HMGB1 protein was only down-regulated in the supernatant but not in the cell lysis. These features of AGDP inducing HMGB1 expression in different scenarios may be related to the alterations of cellular locations of HMGB1, since AGDP treatment could aggravate the accumulations of nucleus HMGB1 protein and thus inhibit the extracellular release of HMGB1 in the context of PRV infection. Although the underlying mechanism for AGDP regulating the output of nuclear HMGB1 remains unclear, it seems likely that AGDP exerting anti-inflammation effects in responding to PRV infection regulates the expression of HMGB1 in vitro.

PRV infection [18] and the RAGE-HMGB1 axis [33] could all induce the inflammatory response by activating the signaling pathway of NF-κB. To further investigate the anti-inflammatory function of AGDP during PRV infection, the effects of AGDP on the activities of IκBα and P65 proteins were further explored, which are thought of as the important checkpoints in the NF-κB signaling pathway. As expected, AGDP could suppress the degradation of IκBα and decrease the phosphorylation levels of P65 in PRV-infected MLE-12 cells. These results suggest that the potential pathway of AGDP in anti-inflammation may be associated with the NF-κB signaling pathway. NF-κB is considered a promising target for developing new anti-inflammatory drugs to treat inflammatory diseases. Although AGDP suppressed PRV replication and inhibited the inflammatory cytokines produced by MLE-12 cells, it shows a therapeutic role in vitro.

In conclusion, the present study provides insight into the pathophysiology process of PR from a peptidomic angle with the successful application of LC-MS/MS. By comparing the peptides in PRV-infected and control groups, this study screened many meaningful DEPs that displayed their roles during PRV infection. In particular, one of the putative functional peptides, AGDP, was synthesized and was found to have a significant therapeutic effect in PRV infection in vitro. These findings provide insight into the search for sensitive and non-invasive biological markers and present novel clues for the designing and developing of more effective therapeutic strategies for PRV infection, which may support the traditional therapies for PRV infection. However, various factors could influence the reliability of our results, including detection methods, enzymatic activity, sample numbers, etc. Thus, further in-detail studies are needed to investigate the roles and mechanisms of AGDP in PRV infection, and these will facilitate the prevention of PRV infection.

## 4. Materials and Methods

### 4.1. Cells, Virus, and Infection

MLE-12 and PK-15 cells were maintained at 37 °C with 5% CO_2_ in Dulbecco’s modified Eagle’s medium (DMEM) supplemented with 10% (*v*/*v*) fetal bovine serum (FBS) (Gibco, MA, USA), 100 units of penicillin G, and 100 µg of streptomycin. The virus (PRV, Min-A strain) was propagated according to the previous report [14]. The obtained virus titration was 4.6 × 10^8^ PFU/mL and stored at −80 °C until for further use. MLE-12 cells’ infection with the PRV was completed under biosafety level 2 (BSL-2) laboratory conditions at a multiplicity of infection (MOI) of 1. After adsorption at 37 °C in 5% CO_2_ for 1 h, the cells were washed with phosphate-buffered saline (PBS) and then added to DMEM cultured for 24 h.

### 4.2. Animals, Virus Infection, and Samples Collection

The animal experiments followed governmental and international guidelines. In the present study, specific pathogen-free (SPF) grade C57BL/6J mice aged at a range of 6 to 7 weeks were obtained from the WUSHI animal center (Shanghai, China). All the mice were reared in sterile cages and placed in an SPF chamber for a 12 h to 12 h light–dark schedule and were free to consume feed and water. Twenty mice weighted at approximately 18–20 g were randomly divided into two groups: group 1 and group 2. Group 1 was employed as the control group, and group 2 was the PRV-infected group. Both groups contained ten mice. For the experimental setting, all animals in group 1 were treated with 100 μL DMEM (+2% FBS), and the animals in group 2 were intramuscularly injected with 100 μL 4.6 × 10^8^ PFU of PRV. All mice were monitored daily for clinical signs and symptoms to 3 days post-infection (dpi). Then, all mice were anesthetized and sacrificed. Tissues from two groups were, respectively, collected and saved for histopathological observation (hematoxylin and eosin, H&E), gene expression (PCR/qPCR and RT-PCR/qRT-PCR), protein expression (Western blotting, WB), and peptidomic analysis with LC-MS/MS.

### 4.3. Histopathological Observation

Tissues (lungs) collected from mice in PRV-infected and control groups were postfixed with 4% paraformaldehyde solution. Then, the tissues were embedded with paraffine, and the paraffine-embedded sections were cut into 5 μm-thick sections with Leica SM2000R rotational microtome (Leica, Nussloch, Germany) according to the previous reports [42]. Tissue sections were deparaffinized in xylene, rehydrated in a grade series of ethanol, and stained with H&E using standard procedures. Then, the sections were observed under a bright-field microscope (Nikon, Tokyo, Japan) and photographed.

### 4.4. Peptidomic Analysis

Firstly, total proteins were extracted from mouse lungs tissue and filtered by 10 KDa ultrafiltration tube, and the obtained penetrating fluid was the peptide sample, to which we added 1% trifluoroacetic acid (TFA) to adjust the pH to 2–3. Then, 100 μg peptide sample was taken for enzyming and desalting and was dried with a vacuum concentrator. We dissolved the peptide samples in 20 μL of dissolution buffer (0.5 M tetraethylammonium bromide, TEAB), added 70 μL of isopropanol, and centrifuged it with shaking. Samples were labeled with iTRAQ Reagent-8 plex Multiplex Kit (AB Sciex, MA, USA) according to the manufacturer’s instructions. Then, the peptide samples were desalted and dried with a vacuum concentrator. Peptide samples were diluted into 1 μg/μL on-board buffer and the volume was set to 8 μL, and they were analyzed using Triple TOF 5600 + LC/MS system (AB Sciex, MA, USA) for mass spectrometry data acquisition. Proteopilot 4.5 software (July 2012; ab SCIEX) was used to retrieve and analyze the mass-spectrometer data. To be considered as being differentially expressed, peptides were required to have a *p* ≤ 0.05 and fold change ≥1.5 (defined as up-regulated) or fold change ≤0.67 (defined as down-regulated). General properties of the DEPs were analyzed by Volcano maps, heat maps, GO, and KEGG bioinformatics analysis.

### 4.5. Peptide Synthesis

One of the potential DEPs, AGDP, was synthesized and purified by the Hangzhou Peptide Company (Hangzhou, China), derived from the precursor protein of RAGE. Then, the synthetic AGDP was dissolved in 0.01 M PBS at doses of 100 μM and stored at −80 °C for subsequent experiments. The working doses of AGDP used here were determined based on our previous experiments.

### 4.6. Effects of AGDP in Responding to PRV Infection In Vitro

To explore the potential effects of AGDP on PRV infection in vitro, MLE-12 cells were infected with PRV. One hour after infection with PRV, the cells were incubated with 100 nM AGDP and cultured at 37 °C in 5% CO_2_. Meanwhile, the mock-treated cells were incubated with DMEM without AGDP. Cells were harvested at 24 h post-infection (hpi) for gene expression analysis by PCR/RT-PCR and qPCR/qRT-PCR, protein expression by WB, and protein localization by immunofluorescence chemistry (IFC). Meanwhile, the supernatants were collected for virus titration.

### 4.7. PCR/RT-PCR and qPCR/qRT-PCR

DNA and total RNA were, respectively, extracted from cell samples and tissue samples using commercial kits (EasyPure DNA kit, Biobase, Shandong, China; TIANGEN Biotech, Beingjing, China). A total of 2 µg RNA was used to synthesize into cDNA with Moloney murine leukemia virus (M-MLV) reverse transcriptase (Promega, WI, USA). Then, the genes mentioned in this study were amplified through PCR and RT-PCR using EasyTaq DNA Polymerase (Biobase, China), qPCR and qRT-PCR using TransStart Tip Green real-time PCR SuperMix (Promega, WI, USA). The amplified products of PCR and RT-PCR were separated on 1.5% agarose gels. The qPCR and qRT-PCR analysis results were shown in normalized ratios, which were auto-calculated using the ΔΔCT method by LightCycler system (Roche, Basel, Switzerland). β-actin was used as the reference housekeeping gene for internal standardization. All the primer sequences for gene application and amplification reactions are listed in the following Table 1.

### 4.8. Western Blotting (WB)

Proteins were extracted from cell and tissue samples and then isolated by SDS-PAGE in 10% polyacrylamide gels. We transferred the isolated protein glue to the nitrocellulose (NC) membrane. Bands were detected using rabbit anti-PRV gE (the antibody was donated by Professor Ma Yanmei at Fujian Agriculture and Forestry University), rabbit anti-HMGB1 (A19529, ABclonal Technology, Hubei, China), rabbit anti-P65 (D14E12, Cell Signaling Technology, MA, USA), rabbit anti-p-P65 (93H1, Cell Signaling Technology, MA, USA), rabbit anti-IκBα (D14E12, Cell Signaling Technology, MA, USA), and rabbit anti-β-actin polyclonal antibody (R019, Transgen biotech, Beijing, China). The secondary antibody for detection used goat anti-rabbit antibodies (131,879, Jackson ImmunoResearch Laboratories, PA, USA). The blots were developed using the FluorChem M Imaging System (ProteinSimple, CA, USA).

### 4.9. IFC and Laser Scanning Confocal Microscopy

The cells were evenly coated on the slides, fixed with 4% paraformaldehyde for 30 min, then the residual formaldehyde solution was washed with distilled water and rinsed with PBS for 5 min. Trypsin repair solution was dropped and soaked for 5 min, then PBS was rinsed for 5 min. We incubated the cells with 5% BSA blocking solution at 37 °C for 30 min. The diluted primary antibody was dropped and incubated at 37 °C for 2 h, then rinsed with PBS 3 times, 5 min each. Fluorescence secondary antibody was dropped and incubated at 37 °C for 30 min, then incubated at 37 °C for 30 min, then rinsed with PBS 3 times, 5 min each. The solution containing DAPI was added, and the coverslip was covered and then observed with a laser confocal microscope (Carl Zeiss LSM 880, Oberkochen, Germany) after standing for 10 min.

### 4.10. Plaque Assay

PK-15 cells were laid in 6-well plates for later use. A suitable gradient dilution (10^−1^ to 10^−8^) of the virus was used, along with maintenance solution (2% FBS+ non-double-resistant medium). After the cells were cleaned twice using PBS, the diluted viruses were added and absorbed for 1 h, and the cells were shaken every 15 min. The supernatants were discarded and washed twice with PBS. A total of 2 mL (2% FBS+ 1% methyl cellulose medium) was added and placed in an incubator for 2–3 days, the numbers of virus plaques were counted, and the corresponding virus titer was calculated.

### 4.11. Statistical Analysis

Statistical analysis was completed by one-way ANOVA using SPSS 19.0 (Chicago, IL, USA) and GraphPad Prism 8.0. Differences were considered significant when *p* < 0.05. Other data are presented as means ± standard deviations (SDs).

## Figures and Tables

**Figure 1 ijms-23-03306-f001:**
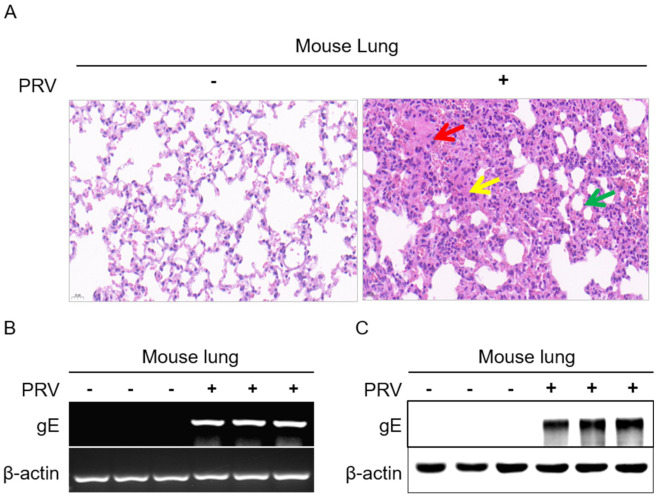
The identification of PRV−infected mice. (**A**) The histopathology of mice lungs was observed by HE staining (40×) results, consolidation of tissue (red arrow), internal hemorrhage of tissue (yellow arrow), markedly enlarged alveolar septa (green arrow); (**B**) The expression of PRV *gE* gene in mouse lung tissues were analyzed by PCR; (**C**) The expression of PRV gE protein in mouse lung tissues were investigated by Western blotting. β-actin was used as reference housekeeping genes for internal standardization.

**Figure 2 ijms-23-03306-f002:**
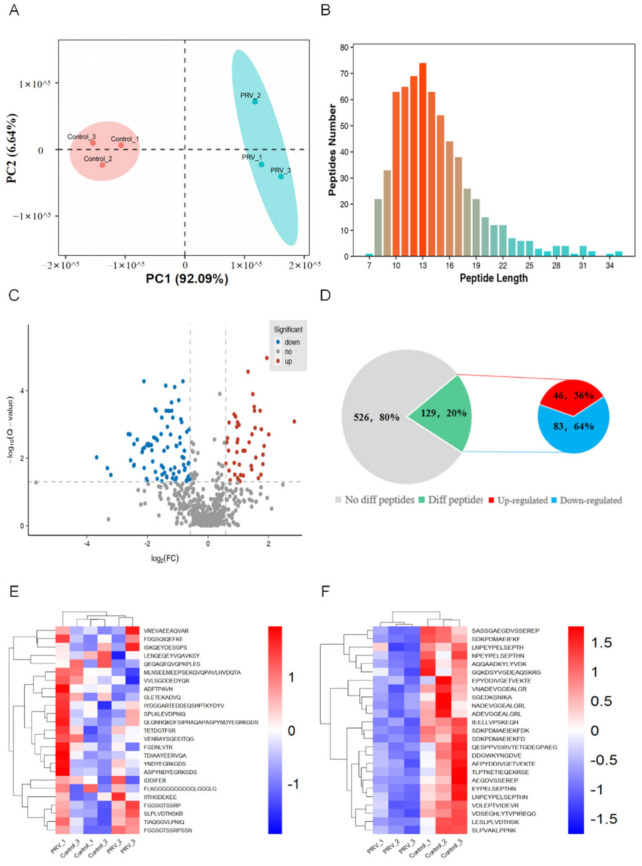
Results of peptidomic data. (**A**) Principal component analysis; (**B**) Map of the distribution of amino acid lengths for all peptides identified by Peptidomic; (**C**) The results of Volcano plot; (**D**) Statistical results of peptidomic; (**E**) Heat map clustering of the top 25 up−regulated peptides; (**F**) Heat map clustering of the top 25 down−regulated peptides.

**Figure 3 ijms-23-03306-f003:**
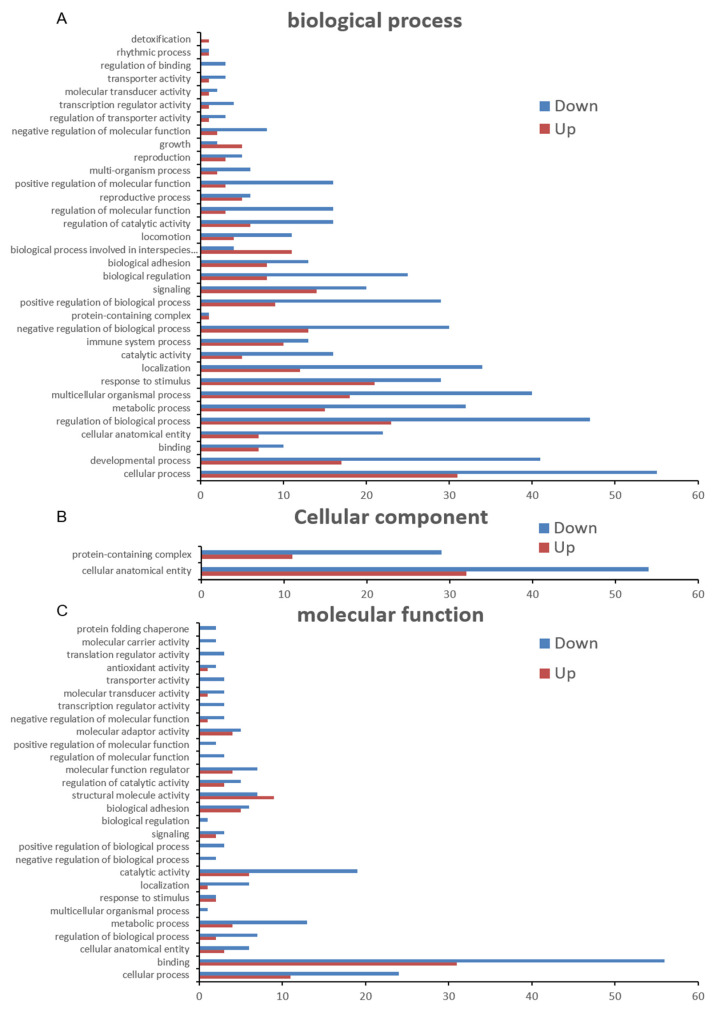
Gene Ontology (GO) annotation for the precursor proteins of the significantly DEPs. (**A**) Biological process; (**B**) Cellular component; (**C**) Molecular function.

**Figure 4 ijms-23-03306-f004:**
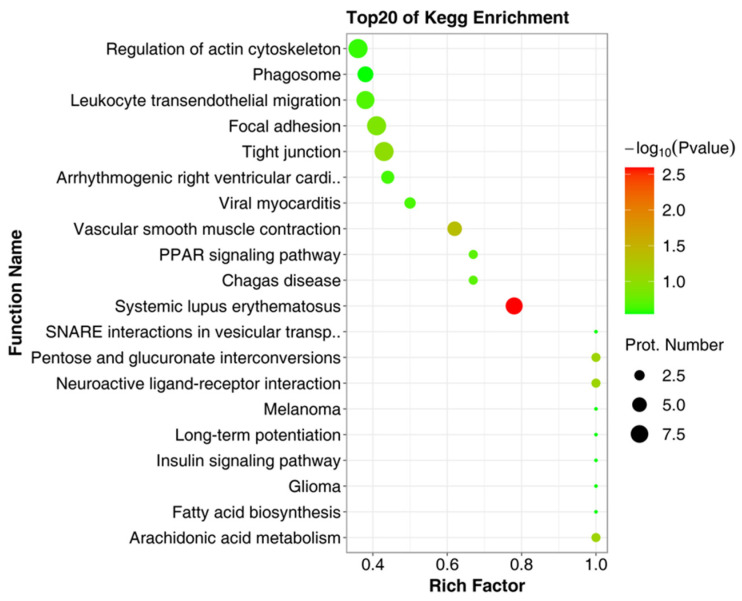
The bubble diagram for the precursor proteins of the significantly DEPs enriched by KEGG.

**Figure 5 ijms-23-03306-f005:**
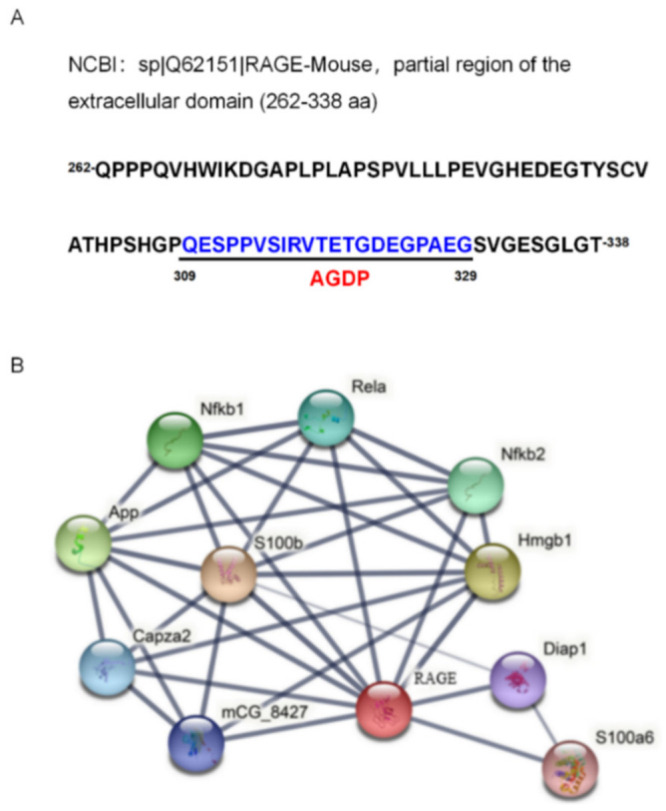
Location of AGDP on precursor protein RAGE and mapping of RAGE protein interaction. (**A**) Localization of AGDP on RAGE protein; (**B**) Interaction network analysis of RAGE and its associated proteins. The image was made according to STRING (Version 11.5, http://string-dg.org/, accessed on 15 December 2021).

**Figure 6 ijms-23-03306-f006:**
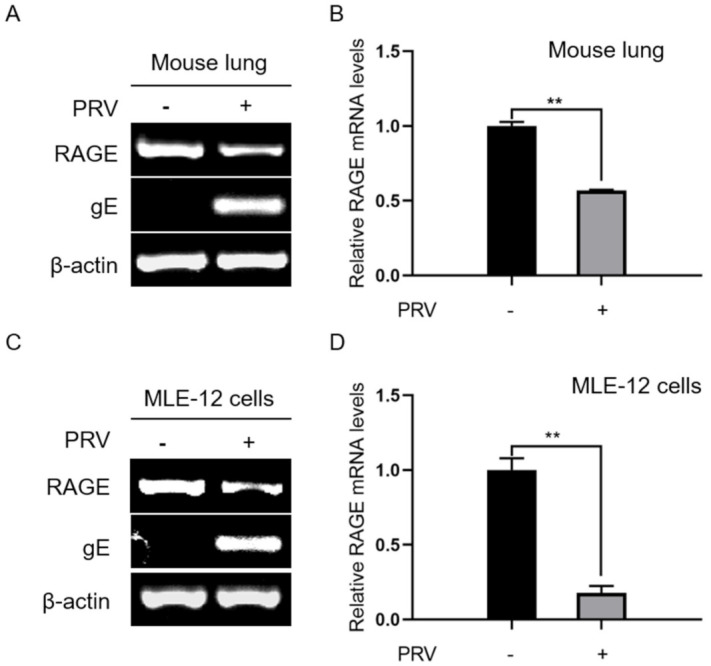
RAGE mRNA expression levels in mouse lungs tissue and MLE−12 cells. (**A**) The mRNA expression levels of RAGE and gE were measured in mouse lungs via RT−PCR at 3 dpi; (**B**) The mRNA expression levels of AGDP were tested in mouse lungs via qRT−PCR at 3 dpi; (**C**) The mRNA expression levels of AGDP and gE were investigated in MLE-12 cells via RT−PCR at 24 hpi; (**D**) The expression levels of RAGE mRNA in MLE-12 cells were analyzed by qRT−PCR at 24 hpi. β−actin was used as reference housekeeping genes for internal standardization. ** *p* < 0.01.

**Figure 7 ijms-23-03306-f007:**
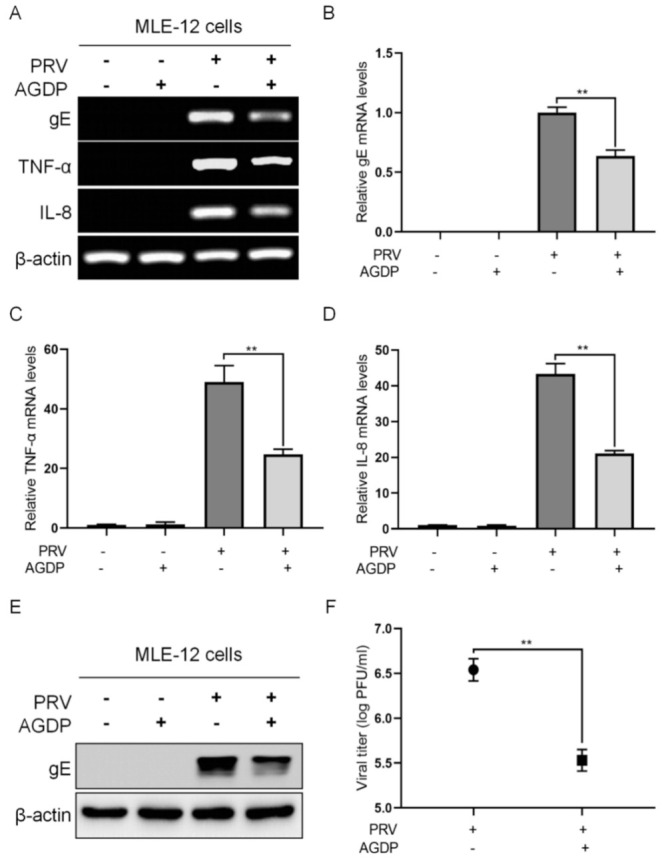
AGDP exerted anti−PRV infection properties in MLE−12 cells at 24 hpi. (**A**) RT−PCR detected the expression of TNF−α and IL−8 mRNAs, and PCR analyzed the expression of gE in responding to PRV infection; (**B**–**D**) qRT−PCR detected the expression of TNF−α and IL−8 mRNAs, and qPCR analyzed the expression of gE in responding to PRV infection; (**E**) The expression level of PRV gE protein in MLE−12 cells were measured by WB; (**F**) The virus titer was detected by plaque assay. β−actin was used as a reference gene for internal standardization. ** *p* < 0.01.

**Figure 8 ijms-23-03306-f008:**
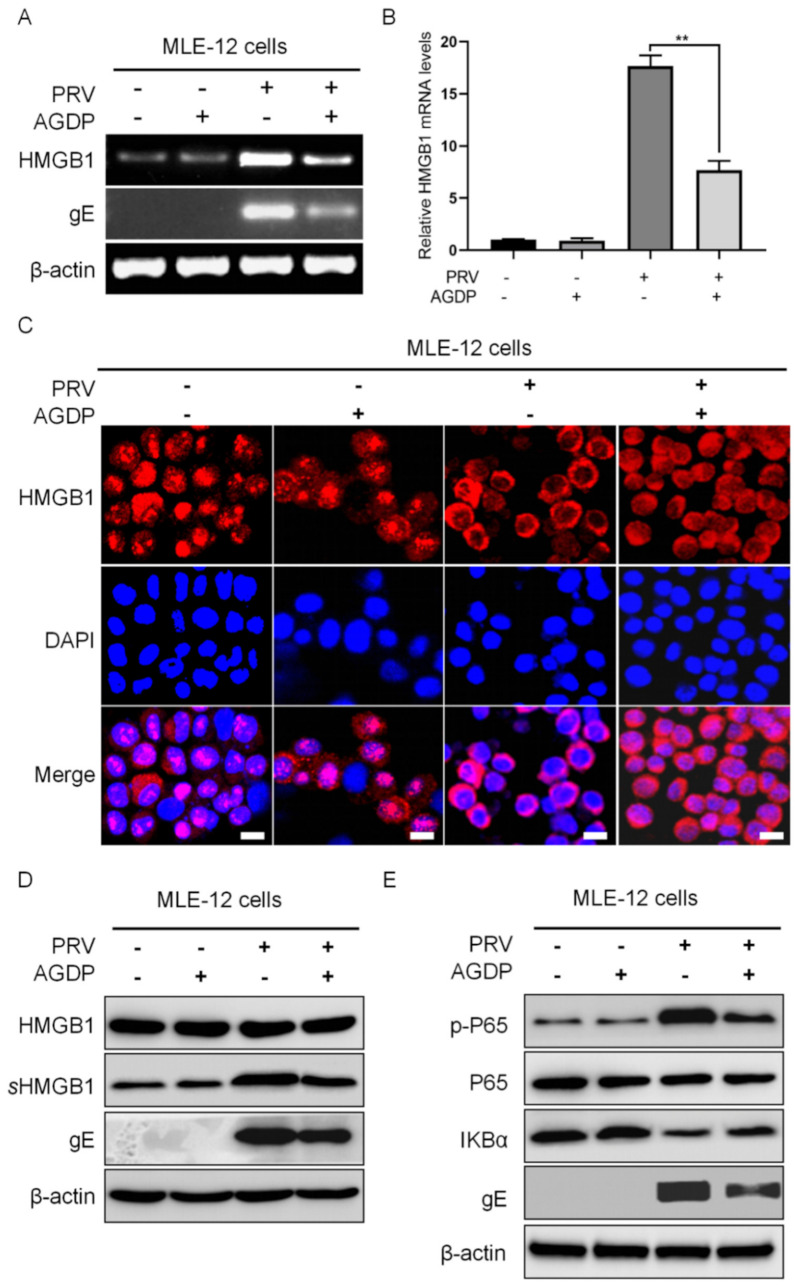
AGDP regulated the activities of HMGB1 in MLE−12 cells under PRV infection at 24 hpi. (**A**) The expression levels of HMGB1 mRNA under the treatment with AGDP were detected by RT−PCR; (**B**) The expression levels of HMGB1 mRNA under the treatment with AGDP were measured by qRT−PCR. (**C**) The alternations of cellular locations of HMGB1 protein induced by AGDP treatment were observed under PRV infection by IFC. scale bars = 10 µm; (**D**) The expression levels of HMGB1 protein were explored under PRV infection by WB; (**E**) AGDP suppressed the degradation of IκBα protein and the phosphorylation levels of P65 protein in PRV−infected cells. β−actin was used as a reference gene for internal standardization. ** *p* < 0.01.

**Table 1 ijms-23-03306-t001:** Primer used for RT-PCR and qRT-PCR.

Names of Primers	Sequences of Primers (5′→3′)
PRV gE Forward	CCACTCGCAGCTCTTCTCG
PRV gE Reverse	CAGTCCAGCGTGGCAGTAAA
Mouse HMGB1 Forward	GCTGACAAGGCTCGTTATGAA
Mouse HMGB1 Reverse	CCTTTGATTTTGGGGCGGTA
Mouse IL-8 Forward	CAAGGCTGGTCCATGCTCC
Mouse IL-8 Reverse	TGCTATCACTTCCTTTCTGTTGC
Mouse TNF-α Forward	ACGGCATGGATCTCAAAGAC
Mouse TNF-α Reverse	CGGCAGAGAGGAGGTTGACT
Mouse β-actin Forward	AATGGGTCAGAAGGACTCCT
Mouse β-actin Reverse	ACGGTTGGCCTTAGGGTTCAG

## Data Availability

The materials in this study are commercially available, except for the rabbit anti-PRV gE antibody.

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
