# Peer review of "Peptidomic Analysis on Mouse Lung Tissue Reveals AGDP as a Potential Bioactive Peptide against Pseudorabies Virus Infection"

_ijms, 2022, doi:10.3390/ijms23063306_

Round 1

Reviewer 1 Report

The authors present their analysis on a pull of peptides caused by a severe histopathological damage during pseudorabies virus infection.

The paper is well written and the analysis is carried out in clear and consistent methodology. 

The introduction is clear and well described and the aims of the manuscript are described exhaustively. The results are clear reported. Discussion is supported by data.

In my opinion, the manuscript has to published in present form. 

Reviewer 2 Report

Ma and colleagues present a peptidomic analysis revealing a AGDP as a possible bioactive peptide against pseudorabies virus (PRV). They identified over 100 peptides in the lungs of mice infected with pseudorabies and investigated the role of one, in particular, AGDP as a possible anti-PRV drug.

Grammar, syntax and punctuation errors are too numerous. Excessive use of etc. and et al”  While I realize the translation from another language can be problematic, the use of an editor is needed before submitting the manuscript for review.

Introduction: Line 30: Sentence should end after Varicellovirus.  The next sentence should start with Its not it’s, which is a contraction for “it is”.  Your use of “its” is possessive like my or your so no apostrophe. Line 32: PRV can infect, not could.  Line 33: Sentence begins with “In particularly”, It should be particular, however you should start that sentence with “Pigs are the only natural host…..”

Line 36: You write of effective measures but only name one: vaccination.

Line 38: recommend using “Specifically” to begin the sentence rather than “importantly” because that links it to the previous sentence, which is rather vague.

Line 39: in 2018, not at 2018.

Line 41: What are These? Are you referring to the variants or the symptoms? Also the term “that” should be used before traditional vaccines.

Line 42:  “Therefore, there” does not work.  I think you mean necessity not necessary.  This sentence also makes little sense and is excessively wordy.

Line 46.  Begin the sentence with “The host”.

Line 47: Suggested sentence: Several studies have demonstrated that the underlying interactions between PRV and the host innate immune system are primarily mediated through several signaling pathways…..and modulate….

Line 69: That is not a sentence.

The introduction is very poorly written (or translated)

Methods

2.4 Spell out your acronyms the first time you use them i.e. TFA and TEAB

Results

Figure 3 is too small to be readable.  I recommend putting each biological process as it’s own figure and labeling them A, B and C

Maybe have the Kegg map as D?

The sentence beginning on line 252 belongs in the discussion.  Also the one beginning on line 255, because it is either hypothesis or conclusion.

Caption for figure 4: The image was drawn….

Section 3.3.  Try to avoid using showed and using demonstrated, revealed or indicated instead.

Sentence beginning on line 269 belong in the discussion.

Section 3.5

Sentence beginning on line 298 belongs in discussion as well as most of this section should be in discussion.  Results are for just stating your results, what the results actually mean should be in the discussion section.

The figures are very nice and you seem to have gotten good results but this paper is so poorly written I would have to recommend a rewrite before publication.

Round 2

Reviewer 2 Report

Thanks for all the edits